# Secondary Impact of COVID-19 Pandemic on People with Parkinson’s Disease—Results of a Polish Online Survey

**DOI:** 10.3390/brainsci12010026

**Published:** 2021-12-26

**Authors:** Karolina Krzysztoń, Beata Mielańczuk-Lubecka, Jakub Stolarski, Anna Poznańska, Katarzyna Kępczyńska, Agata Zdrowowicz, Izabela Domitrz, Jan Kochanowski

**Affiliations:** 1Department of Neurology, Faculty of Medical Sciences, Medical University of Warsaw, 01-809 Warsaw, Poland; beata.lubecka@wum.edu.pl (B.M.-L.); jakub.stolarski@wum.edu.pl (J.S.); kkepczynska@wum.edu.pl (K.K.); azdrowowicz@wum.edu.pl (A.Z.); idomitrz@wum.edu.pl (I.D.); jan.kochanowski@wum.edu.pl (J.K.); 2Department of Population Health Monitoring and Analysis, National Institute of Public Health NIH–National Research Institute, 00-791 Warsaw, Poland; apoznanska@pzh.gov.pl

**Keywords:** Parkinson’s disease, COVID-19, pandemic

## Abstract

The COVID-19 pandemic causes increased mental stress and decreased mobility, which may affect people with Parkinson’s disease (PD). The study aimed to investigate the secondary impact of the COVID-19 pandemic on the level of activity, quality of life (QoL) and PD-related symptoms. The respondents completed an online survey in Polish in the period from December, 2020 to June, 2021. The questionnaire was completed by 47 participants aged 43 to 90 years (mean 72.1 ± 1.3 years). A total of 94% reported reduced contact with family and friends. Over 90% remained active during the pandemic. However, 55% of people with PD showed subjectively lower level of activity then before the pandemic. Moreover, 36% of the respondents felt afraid to visit a doctor and reported problems with access to medication. Subjective QoL reduction was reported by 80%, and 83% declared worsening of PD symptoms. The post pandemic deterioration of motor symptoms in people with PD did not affect their QoL. However, the deterioration of contacts and feelings of isolation had a significant impact on the decline in quality of life (*p* = 0.022 and *p* = 0.009, respectively) and the presence of anxiety (*p* = 0.035 and *p* = 0.007, respectively). These results may indicate than greater importance of social and mental factors than fitness and health-related factors in the QoL self-assessment of the people with PD.

## 1. Introduction

Social distancing, isolation, increased psychological stress and reduction in mobility related to the COVID-19 pandemic have affected the lifestyle of people around the world [1,2]. The COVID-19 pandemic contributed to changing the management of patients with movement disorders [3,4]. Access to physicians and physiotherapists was restricted. In response to the situation, the role of telemedicine (used in general to diagnose, consult and heal, also in the meaning of telerehabilitation) increased [5,6,7] and its effectiveness was confirmed even before the pandemic [8,9,10].

It has been hypothesized that people with Parkinson’s disease (PD) may be at an increased risk of developing COVID-19 [2,11,12] due to an indirect relationship with age, multiple comorbidities and the frequency of polypharmacy. Respiratory disorders constitute a direct risk factor for more the severe respiratory complications of COVID-19 infection in this population [12,13]. The circumstances of the pandemic may indirectly affect the motor and non-motor symptoms of Parkinson’s disease, leading to the exacerbation of disease symptoms [13]. The effect of the COVID-19 pandemic on people with Parkinson’s disease is the subject of research in many countries [14,15,16]. However, we found no reports for Eastern Europe regarding the time when such research was conducted. Moreover, most of the available reports were peeformed in the early pandemic period. Nevertheless, the model of care for people with Parkinson’s disease still requires improvement in Poland. There is a lack of comprehensive care and, above all, funding from the National Health Fund. Apart from patient clinics and associations or foundations, there is no specialist rehabilitation program and advanced treatment options are under development, which means that their availability for patients is still limited. We investigated if the secondary impact of the pandemic on people with PD is similar to or different from that in Western countries, where comprehensive care programs are already implemented and where access to treatment, including specialist rehabilitation, is wider and easier.

The first case of coronavirus infection in Poland was confirmed by the Ministry of Health on March 4, 2020. Therefore, with the epidemic spreading around the world, the first restrictions were introduced, and, on 16 March 2020, the state of epidemiological emergency was introduced in Poland. In October, the epidemic situation in Poland considerably deteriorated, and daily counts of new infection cases reached the tens of thousands (data of the Polish Ministry of Health, December 2020). However, at a time when the pandemic was already taking a terrible toll in the Western countries, there were not many cases in Central and Eastern Europe and there was practically no first wave of the pandemic in Poland. Therefore, it seemed appropriate to investigate the impact of the pandemic for a longer period. Moreover, this study was conducted once vaccine development was complete. This could be related to many different feelings. On the one hand, it was a step to stop the pandemic, the step that many people were waiting for. However, people could be afraid of a new, as yet unknown vaccine, not knowing about the possible interactions with their underlying disease. In addition, the availability of vaccines for the entire population was uncertain at that time. Those factors could additionally influence the participants’ different points of view. Thus, it was decided to assess the situation at the time of the introduction of the vaccine to the market. One additional motive for creating this survey was related to a need to become acquainted with the situation of people with Parkinson’s disease and its subjective assessment while waiting for the developed vaccine (completion of a certain stage, initiation of the next one), which potentially allows for further observation of this group of patients at subsequent stages of the pandemic, or rather at subsequent stages of the fight against the pandemic.

An online survey was designed and used in order to investigate the secondary impact of the COVID-19 pandemic on the quality of life and the activity level of people with PD in the Polish population.

## 2. Materials and Methods

Individuals with PD participating in the online study were invited to complete an online survey between 23 December 2020 and 23 June 2021. We used this research method in the pandemic period hoping to get data without the need of meeting the participants. The invitation was sent by email to the main associations of people with Parkinson’s disease[note 1]. Patients from the Neurological Clinic were also included into the study after they had agreed to be added to the mailing list. The people with PD were allowed to complete the survey anonymously by themselves or with the help of their caregivers.

The survey was prepared in Polish after reviewing the results and research questions of recent survey studies concerning the impact of the pandemic on people with PD conducted in other countries [17,18]. Before filling out the questionnaire, the participants were informed that they expressed their consent to participate in this study by sending it back. The main part of the survey consisted of questions related to COVID-19 status and associated outcomes (based on part of the Spanish questionnaire [17]), fear concerning the pandemic, physical activity, changes in daily activities and actions, and changes in PD symptoms (motor and non-motor, including depression and depressed mood). The last part referred to the course of PD (duration, level of independence) and basic demographic characteristics, including sex, age, level of education and the place of residence (the complete questionnaire is available in the Appendix A). After the complete survey was prepared, it was reviewed by two people with PD and a caregiver to verify the readability.

The email contained an undefined link to the survey (i.e., the links were not specific to the email recipient). Due to the inability to track the respondents of the survey (the answers were anonymous), no reminders concerning the surveys were sent.

After completing the survey, the participants received a link to an approximately 30-min video of exercises prepared by the authors especially for people with PD, performed by an experienced physiotherapist. This video acted as a kind of incentive for the respondents. An additional goal was to motivate the participants to exercise at home daily, using the presented set of exercises, as stated in the invitation to the study.

An automatic method for capturing only complete responses with the use of Google forms was selected.

The study design followed the recommendations of good practice in survey studies by Kelley et al. [19] and the checklist for reporting results of Internet e-surveys [20].

### Statistical Methods

Survey results were summarized using descriptive statistics. Percentages were used for categorical variables, the mean value with standard error and median were used for age, and quartiles were used for values measured with the 1–5-point scale, i.e., activity level and quality of life. The limits of 95% confidence intervals (95% CI) for the mean and fractions were calculated.

The results of one open question were analyzed. The answers to the open question concerning beliefs about threats from the SARS-CoV-2 virus were analyzed independently by three researchers, and the conclusions were discussed. The most common terms were identified (high mortality/high infectivity/severe disease), coded as dummy variables, and included into the statistical analysis.

The distribution of answers to all questions is provided in the Appendix A. Most of the questions had three categories of answers: YES, NO, DO NOT KNOW (or equivalent), and the percentages were found to be sufficient. With more responses, the data were compiled in tables or figures presented below and in The Appendix A.

The differences in the distribution of continuous variables were evaluated using the Mann–Whitney test for 2 compared groups and the Kruskal–Wallis test followed by the Dunn post-hoc test for 3 groups. The chi-square or the Fisher exact tests were applied to compare categorical variables adequately to the number of expected cases. The significance level of 0.05 was assumed. However, some results on the border of significance (*p* < 0.10) are also presented. The analyses were performed using the SPSS12.PL software package.

## 3. Results

### 3.1. Participants/Demographics

A total of 47 participants completed the survey, 10 by themselves and 37 with the help of caregivers.

The patients’ age ranged from 43 to 90 years, with the mean being 72.1 ± 1.3 years (mean standard error) and the median being 71 years. The majority of the respondents were male (30/47—64%). Detailed data concerning the study participants (group characteristics) are presented in Table 1.

### 3.2. Impact of COVID-19 Pandemic on PD—General Findings

About 55% of the respondents felt particularly vulnerable to contracting COVID-19. Almost all of them (96%) considered the virus particularly dangerous, pointing to high infectivity (47%), severe disease (38%) and high mortality (19%)—detailed information is presented in Figure 1. Fewer than 43% of the respondents had tried to obtain some information about the interaction of SARS-CoV-2 virus with Parkinson’s disease. A total of 53% of them used the Internet, 11% looked for information on the association’s website, and 4% asked the family or friends. No one used the knowledge of medical staff.

Only 6% of the respondents declared that they had not changed their behavior since the virus was identified. Out of 47 people, 46 (98% of total) followed COVID-19 protection measures. The most common measures used are presented in Figure 2.

More than half of the respondents (51%) were not in contact with a SARS-CoV-2 infected person, 32% of them reported to have a contact and 17% did not know if they had.

Over a third of the respondents (36%) were still afraid to visit their doctor and another 30% had that problem for some time during the pandemic. Moreover, 36% reported having a problem with access to medications that were constantly taken. More than half of the respondents had problems with access to rehabilitation during the pandemic (including 11% of people who reported having problems for some time during the pandemic). More than 90% of the subjects were active (exercising)—43% exercised under the supervision of a specialist (physiotherapist, instructor or speech therapist), and only 21% used online visits. Detailed patient physical activity data are shown in Table 2.

As regards the question “How would you rate your activity before the pandemic (1) and now (2) with a score from 1 to 5?”, a total of 55% of people with PD indicated a lower level of activity (in comparison with the state before the pandemic), 30% found no difference and a positive difference was reported by 15%. The exact distribution is shown in Appendix A.

Moreover, 70% of the respondents stated that the COVID-19 pandemic negatively affected their fitness.

The deterioration of Parkinson’s disease symptoms was reported by 39 people (82% of all respondents). They listed the symptoms presented in Table 3.

As many as 94% of respondents reported deteriorating contact with family and friends. When asked about the feeling of anxiety during the pandemic, 38% answered in the affirmative, 34% indicated “maybe”, and 28% did not report such feelings, respectively. Answers about feelings of anxiety are shown in Figure 3. The distribution of answers to the analogous question concerning the feeling of isolation was similar: 41%, 36%, and 23%.

As regards differences in the quality of life (QoL) before and during the pandemic (Figure 4), most of the respondents reported a decrease of 1 (47%), 2 (19%) or 3 (4%) points (respondents pointed from 1 meaning very poor to 5 meaning very good QoL, pre-pandemic and at the moment of completing the survey). The average change of QoL (median) decreased by 1 point. Moreover, 28% of respondents did not report any change, and 2% stated that their QoL improved.

### 3.3. Impact of the COVID-19 Pandemic on the Worsening of PD Symptoms

The frequency of reporting the exacerbation of disease symptoms was significantly associated with the sex of the respondents—it was declared by all surveyed women and 73% of men (*p* = 0.038). The deterioration of symptoms was not significantly related to other sociodemographic characteristics (including age), or to disease duration. All respondents who were completely dependent reported deterioration, but the difference from other respondents was not statistically significant (100% vs. 79%). The exacerbation of PD symptoms was reported by all people who had been in contact with COVID-19 patients and by 75% of others (*p* = 0.042). Those who believed that SARS-CoV-2 virus was particularly dangerous because of the severity of the disease it caused experienced a deterioration more commonly (94% vs. 71%, *p* = 0.088, the difference was on the border of statistical significance).

People who experienced the deterioration of disease symptoms decreased their activity during the pandemic much more often than others (62% vs. 25%). Among the patients who exercised with a specialist online, the deterioration was reported by 60%, and among those who did not exercise in this way deterioration was reported by 89% (*p* = 0.051, the difference was on the border of statistical significance). Moreover, the patients with the exacerbation of disease symptoms were characterized by significantly lower activity than others at the time of completing the questionnaire. The median was on average 2 vs. 3 points (*p* = 0.095, the difference was on the border of statistical significance). The exacerbation of disease symptoms was statistically significantly associated with deterioration of fitness, as reported by 77% of people with severe of PD symptoms and 38% of the others (*p* = 0.040).

We also made two important observations (clear, but statistically insignificant due to the small size of the studied group) concerning the deterioration:The respondents who experienced interruptions in access to PD-related medicines were also more likely to experience deteriorating symptoms (94% vs. 77%).People who sought exercise on their own reported deterioration much less often (77% vs. 91%).

The severity of mental symptoms (depression, mood or memory problems) was statistically significantly associated with the feeling of isolation (it concerned 6% of the respondents who did not feel isolated and 40% of the others, *p* = 0.017).

### 3.4. Impact of COVID-19 Pandemic on the Activity Level in PD

The identified associations between physical activity during the pandemic and the severity of Parkinson’s disease symptoms are presented above.

Women significantly more often than men decreased their activity (76% vs. 43%, *p* = 0.028). The distribution of activity change significantly differed in patients from the clinic and in the others (*p* = 0.030). On average, the patients from the clinic decreased their activity by 2 points (the median), while the remaining patients did not change their level of activity (Figure 5a). People from the clinic were older (average age 75 vs. 70 years), but the age difference was not statistically significant.

Contact with a COVID patient (in the 3-group approach: no/yes/do not know) showed a significant statistical relationship with the change in activity (*p* = 0.046), as shown in Figure 5b. The activities of people who did not know whether they had been in contact decreased the most—by 2 points on average. People who had been in no contact decreased their activity by one point, while those who had been in contact did not change their activity (the median change = 0, statistically significant difference in relation to the former group, *p* = 0.020).

People who decreased their activity significantly more often declared that the pandemic negatively affected their fitness (92% vs. 43%, *p* <0.001). On average, those who were losing fitness decreased their activity by one point (the median), while those who kept fitness did not change its level (*p* = 0.003).

### 3.5. Impact of the COVID-19 Pandemic on the Quality of Life

In this section, we identify factors associated with the change in the quality of life during the pandemic.

People suffering from disease for more than 5 years experienced a significantly greater decrease in the quality of life than patients with a shorter history of disease (*p* = 0.047). Despite equal medians (in both groups the average quality change equaled −1), the distributions were different. A one-point decrease or no change in the quality of life was typical (within an interquartile range) in patients with shorter PD duration, while among the others the decrease was of 1 or 2 points (Figure 6a). People with longer PD experience were on average older (median age 69 vs. 73.5 years, *p* = 0.011).

There was a statistically significant association between the feeling of isolation and the change in the quality of life (*p* = 0.009; Figure 6b), people feeling isolation typically reported a decrease in the quality of life by 1 or 2 points, while the others reported a decrease of 1 point or no changes. The same applies to the frequency of deterioration in the quality of life (observed in 90% of respondents with a feeling of isolation and 57% of the others (*p* = 0.017).

The deterioration of social contacts with the family/friends exhibited a statistically significant association with changes in the quality of life. The respondents who did not report any deterioration in their social contacts did not experience a decrease in the quality of life, which was reported by 75% of the remaining respondents (*p* = 0.022). Regarding people whose social contacts deteriorated, the average change was reported as a decrease by 1 point (median = −1).

All people who underwent strict isolation experienced a deterioration in the quality of life, among the others who did not apply this precaution, this percentage was 62% (the difference was statistically significant *p* = 0.022).

The difficulties with access to medications worsened the quality of life, with the difference being on the border of statistical significance (*p* = 0.090). People reporting problems with medications were on average younger (median age 69 vs. 73.5 years, *p* = 0.014).

The association between the decline in activity and the change in QoL was at the border of statistical significance (*p* = 0.098). On average, those who did not decrease their activity did not report a change in quality of life (median change = 0), while the others reduced their QoL by one point.

### 3.6. Impact of COVID-19 Pandemic on the Anxiety

The frequency of the feeling of anxiety among the respondents showed a statistically significant association with sex and duration of the disease. Its complete absence was more often declared by men (47% vs. 12%, *p* = 0.015). Patients with a shorter disease duration experienced anxiety more often (65% vs. 22%, *p* = 0.005).

The declaration of not feeling anxiety was related to:Lack of conviction about the severity of COVID (45% vs. 17%, *p* = 0.048).Lack of a feeling of isolation (59% vs. 20%, *p* = 0.007).Maintaining social contacts (100% vs. 30%, *p* = 0.035).

It is worth mentioning that the association between interpersonal relationships and the feeling of isolation was not statistically significant.

Respondents experiencing anxiety more often than others avoided visiting a doctor or a patient association (72% vs. 14%, *p* < 0.001) and sought information on interactions between PD and COVID-19 on the Internet (19% declaring no anxiety and 55% others, *p* = 0.018).

### 3.7. Impact of COVID-19 Pandemic on People with PD– Summary

After a thorough analysis of the obtained results, we proposed a possible chain of dependence of the changes in the quality of life during the COVID-19 pandemic on other studied features (Figure 7).

### 3.8. COVID-19 Infection in PD

COVID-19 diagnoses were reported by 3 out of 47 people with PD (2 definite—confirmed by tests, 1 probable, based on the symptoms and contact). One of them stated that the infection affected PD-related symptoms of tremors, stiffness, balance disorders and easy fatigability.

## 4. Discussion

People with Parkinson’s disease are at risk of developing certain cognitive and mental dysfunctions, which may additionally affect (most often enhance) their level of anxiety, depression or other feelings in pandemic circumstances [2,13,21]. In this study, even with such a small sample, the results emphasized the primacy of psychological factors in restoring the level of the quality of life, as opposed to the level of activity or symptoms related to PD, which, in many cases, was also influenced by the pandemic. Such mental disorders may be attributed to the worsening of existing anxiety, the insecurity associated with taking medications during lockdown, and the perceived greater risk of contracting SARS-CoV-2 because of age and chronic condition [22]. Conversely, potential loneliness caused by restrictions may pose particular risks for people with PD, who, as mentioned above, are at a high risk of developing depression. As shown in current study the feeling of anxiety was associated with many factors characteristic of the pandemic period (Figure 7). The obtained sex-dependent associations, in terms of the quality of life, worsening of PD-related symptoms and feeling anxiety, should also be emphasized (despite the small sample size and the higher prevalence of male sex in PD population).

Previous studies [23] showed that people with PD who felt lonely and/or isolated were characterized by greater symptom severity and poorer quality of life. The same correlation was found in our study. Moreover, 26% of our respondents indicated depression or mood disturbances as symptoms that worsened during the pandemic period. However, Luis-Martínez et al. conducted a study in Italy and reported that PD participants who were regularly observed at the clinic and participated in some educational activities reported no difference in the QoL before and during the pandemic [24]. Quite contrary to the results obtained for the Polish population where most of the respondents reported a decrease of the subjective QoL level—which could also be related to the lack of comprehensive specialist care in Poland and indicates the need to create interdisciplinary teams throughout the country. This problem may also be indicated by the fact that none of the respondents tried to inquire about the possible interactions of their disease with COVID-19 from healthcare professionals. A study carried out in China [21] revealed that patients who were assessed using the “39-item Parkinson’s Disease Questionnaire” (PDQ-39) during and after restrictions had significantly different overall PDQ-39 scores, indicating that the quality of life of people with PD was significantly reduced during the prevention period of the pandemic. The quality of life was even poorer when patients had some difficulty visiting the doctor to seek advice or to change their medications [21]. Another global problem was associated with the fact that the COVID-19 outbreak disrupted regular visits by people with PD, including their travel plans and transportation methods [16,25]. Disease progression often means that PD patient need changes in their treatment and, currently, the main treatment for PD symptoms is drug therapy, which means that patients with the disease need regular medication to keep their symptoms under control. As shown in the present study, more than 1/3 of the participants had some difficulty accessing medication, and, interestingly, they were significantly younger than the remaining respondents. Such impediments could be related to changes in activities and the subjective deterioration of symptoms [18,26,27]. However, despite significant percentage differences, these relationships remained only on the border of statistical significance in this study.

The COVID-19 pandemic affects people with PD in many ways. Due to the pandemic and restrictions, some treatments have been unavailable, such as physiotherapy or aerobic exercises, which may explain the subjective feeling of the deterioration of motor and non-motor symptoms [28], as was shown in the present study.

Various reports have been published on the impact of the pandemic on physical exercise—some studies revealed no change in physical activity [29,30] and others showed a decrease [12,31,32]. This study showed that more than half of patients reported a significant reduction in physical activity levels compared to their state before the pandemic. However, because of the research methodology being was only subjective patient assessment, our results were similar to findings by Song J. et al. who concluded that “The COVID-19 pandemic had a clear impact on exercise and subjective symptoms in people with PD, with reduced exercise being related to a subjective increase in both motor and non-motor symptoms of PD” [33]. Conversely, up to 72.7% of the Spanish population reported to have stayed active [17]. In this study, over 90% people were active (exchanging activities), which does not exclude that the level of activity of many people (especially activity related to treatment or rehabilitation in the fields of movement or speech) decreased. In addition, the decline in activity was associated with the declared decline in fitness.

Interestingly, changes in activity did not significantly affect the subjectively perceived quality of life. It is possible that the people who wanted to remain active, continued to exercise at home or elsewhere. Several people also increased their level of activity.

Patient associations play an important role in improving the quality of life of people with PD and their families. Associations provide information, allowing patients to understand the disease characteristics and to educate themselves and their fellows. In Poland, such associations also create special conditions to improve PD-related symptoms, e.g., rehabilitation meetings tailored to the needs of physiotherapy, psychology and speech therapy. In the present study, the results showed that if people with PD were to seek information about the disease and COVID-19 interactions, they did it more often via patient associations than from medical staff (no respondent reported asking medical staff, as mentioned above). Moreover, for the scientific world, these organizations also play a large role in bringing people with Parkinson’s disease in one place and facilitating certain research (e.g., the current results) and vice versa, they enable people to expand their knowledge about relevant science by organizing lectures and meetings with experts. Regrettably, many associations limited their activities during the pandemic, and, at the same time, many people (37% in our study) were afraid to visit them.

The Fox Insight study, a large sample multicenter study, showed that 43% of PD respondents without a COVID-19 infection reported worsening of at least one existing motor symptom and 18% reported that a new PD-related motor symptom occurred. Moreover, 52% experienced worsening of non-motor symptoms (mood, sleep, cognition, etc.) [18]. Over half of the patients of a Spanish study [17] reported that their PD-related symptoms had worsened. A telephonic review study performed in a large sample in Italy [34], revealed that below 30% of the participants noticed any worsening of motor symptoms, mood or anxiety. Therefore, it has been suggested that, overall, people with PD did not experience a subjective worsening of symptoms during the lockdown. These results could have been related to a short lockdown period. In the results obtained in our study, only 28% of PD respondents did not confirm the deterioration of symptoms. Conversely, over 40% of respondents of a smaller sample from India reported worsening of PD-related symptoms after 3 weeks of restrictions [35]. Moreover, 40% of people with PD showed an intensification of motor symptoms in an observational study conducted with the use of the MDS-UPDRS III in Tokyo [36]. In contrast, a study by Guo et al. revealed that almost 80% of patients reported new or worsening PD symptoms during the pandemic, and the change was significantly more severe than at baseline (pre-pandemic period). It is likely that the long-term depression of mood due to the COVID-19 pandemic affected the psyche of people with PD, thereby contributing to further disease progression and subjective worsening of symptoms. In the present study, the severity of non-motor symptoms was related to a feeling of isolation. In addition, an overall worsening of symptoms was associated with a decline in fitness.

Previous studies showed the fear of contracting SARS-CoV-2 infection as the major concern in the PD population [22,27,29,31,37]. Our study also showed that more than half of the respondents confirmed being afraid of infection. Fear may also be related to previous data concerning higher mortality rates in the PD population compared to the general population, especially in those who were older, with longer PD duration, hypertension or coexisting dementia [34,38,39]. However, other scientific studies showed that the risk of infection, severe disease or death among people with PD was comparable to that of the general population [40,41,42]. Interestingly, Yu et al. [43] found that people with Parkinson’s disease have a higher risk of SARS-CoV-2 infection, but not mortality from COVID-19. In our population, fear was confirmed by more than 50% of patients.

Telemedicine is used more and more commonly as part of multidisciplinary care for people with PD. Its importance has significantly increased in the era of the pandemic. Numerous studies indicated a high level of satisfaction with this form, its high efficiency, and cost-effectiveness. Overall, it was mostly well received [6,7,8,44]. Therefore, we expected more responses to the online questionnaire, given that no contact required, possibility of completing the questionnaire at any time of the day existed and a reward of a set of exercises was offered. The video was only available via the link received and at the time of writing this article it had over 400 views, which may indicate a good reception of the exercises by people who decided to complete the survey and return to the presented video multiple times. Of those who remained active during the pandemic, 34% benefited from self-found exercises, and 7 people (15% of the total) benefited from professional online training/physiotherapy and home physiotherapy. Moreover, the group of people training online less frequently (on the border of statistical significance) reported the deterioration of disease symptoms.

The study has several limitations. First of all, the sample is small, which is a potential source of bias. The exact response rate cannot be calculated due to the lack of data on the active email addresses of people with PD. So far, we have not had a register of people with Parkinson’s Disease in our country, which would undoubtedly allow research on larger samples. One important consequence of the small sample is the low power of the performed the statistical tests, which may make it impossible to identify potentially significant associations. However, despite the small study sample, significant results were found in the statistical analysis. The way in which subjects were recruited for the study resulted in selection bias (e.g., almost 60% of the participants completed higher education). Given that it was an electronic survey, it is likely that the technology and email requirements limited its availability to those with better technological skills or access to the necessary technology. However, age was not a determinant (e.g., we had a 90-year-old respondent), and older or less independent patients benefited from the help of a caregiver. The low number of COVID-19 patients in this study made it insufficient to assess its consequences in people with PD in Poland. Moreover, longer follow-up periods are probably necessary to confirm our observations concerning the effects of lockdown on the motor and non-motor symptoms of PD.

In addition, the COVID-19 pandemic is a dynamic and ever-changing situation, so data collected from the respondents may have been affected by changes while the survey was still active. In addition, the survey was made available after the work on the development of the COVID-19 vaccine was completed.

## 5. Conclusions

The results of the study highlight the impact of the COVID-19 pandemic on the physical activity, quality of life and mental state of people with PD, which emphasizes the importance of managing such problems and ensuring comprehensive care for PD patients, including the use of telemedicine.

Although the vast majority of respondents remained active during the pandemic, more than half of the respondents reported a decline in activity levels. The changes were also apparent in the sense of disease progression, in terms of the worsening of Parkinson’s disease symptoms. Additionally, subjective deterioration in mental symptoms was related to the feeling of isolation.

Interestingly, it appears that the quality of life was not affected by either the activity level after the pandemic (or its changes) or the worsening of motor symptoms. Conversely, the deterioration of contacts and the feeling of isolation had a significant impact on the decrease in the QoL—respondents who reported such feelings during the pandemic experienced a decrease in the quality of life significantly more often than others, and the decrease was significantly greater. It may indicate a greater importance of social and mental factors than fitness and health-related ones in the self-assessment of the QoL of people with PD.

Moreover, the results showed some differences between sexes in terms of the quality of life, worsening of PD-related symptoms and feeling anxiety.

## Figures and Tables

**Figure 1 brainsci-12-00026-f001:**
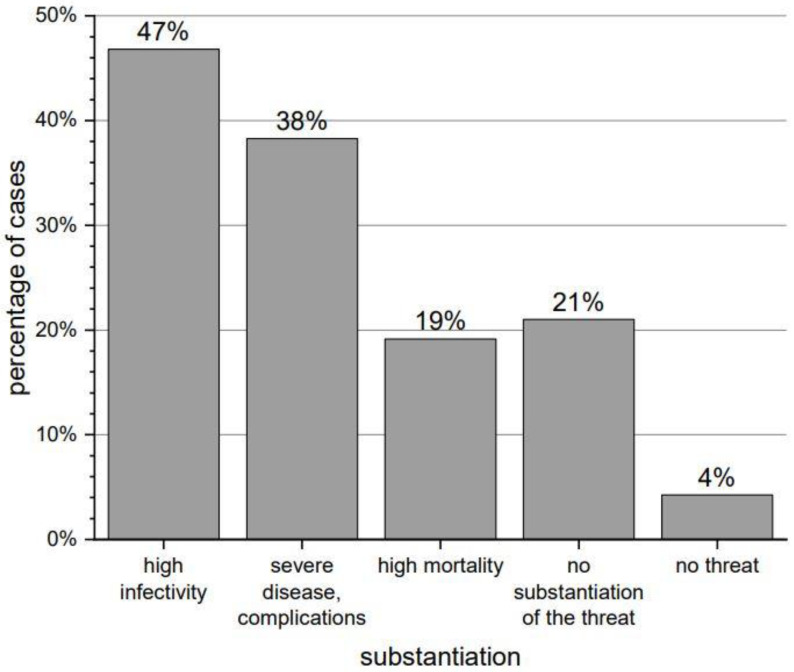
The most frequently quoted substantiation of why the virus is particularly dangerous.

**Figure 2 brainsci-12-00026-f002:**
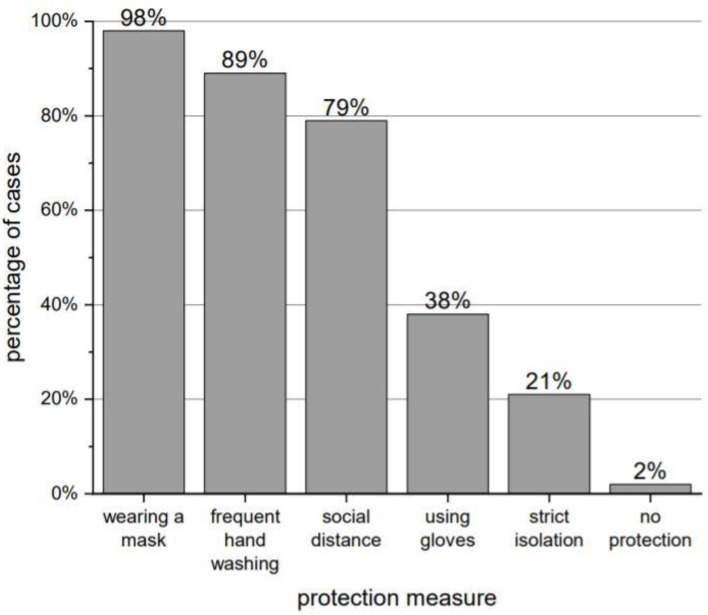
The most commonly used protection to prevent SARS-CoV-2 infection.

**Figure 3 brainsci-12-00026-f003:**
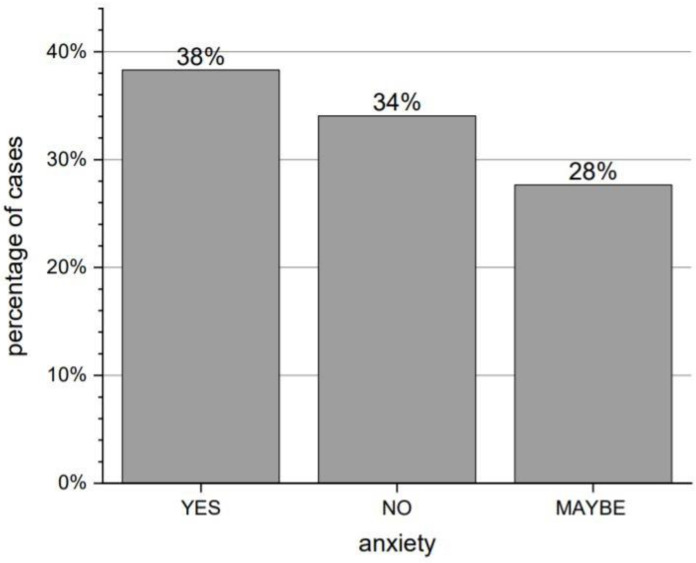
Distribution of answers to the question “Have you felt anxious since the beginning of the pandemic?”

**Figure 4 brainsci-12-00026-f004:**
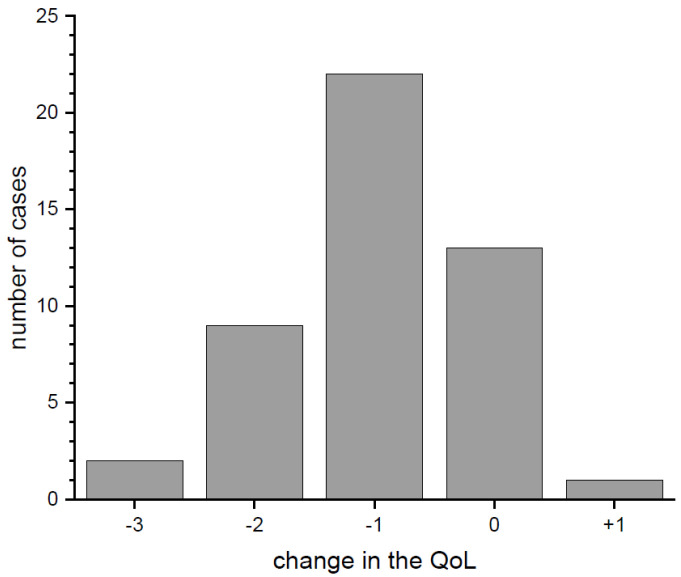
Changes in the level of the quality of life during the pandemic.

**Figure 5 brainsci-12-00026-f005:**
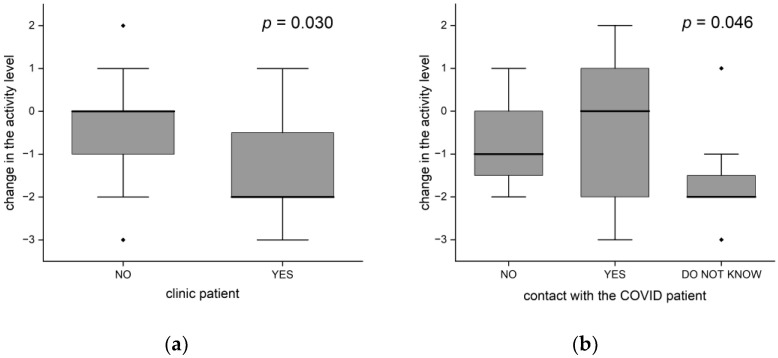
Changes in the level of activity (**a**) differences between clinic patients and others; (**b**) depends on exposure to COVID-19 (the black dots-outliers).

**Figure 6 brainsci-12-00026-f006:**
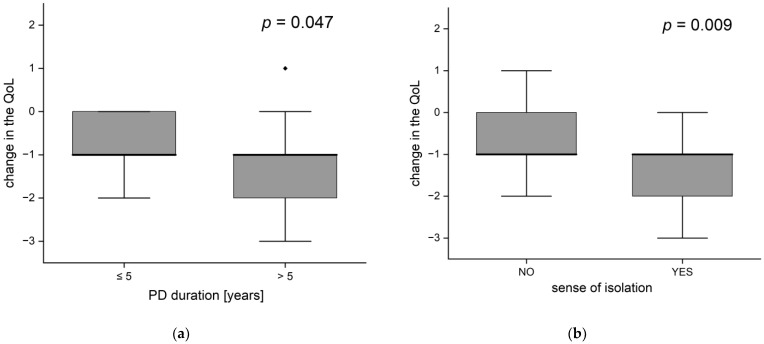
Change of life quality according to (**a**) disease duration and (**b**) the occurrence of feelings of isolation (the black dots-outliers).

**Figure 7 brainsci-12-00026-f007:**
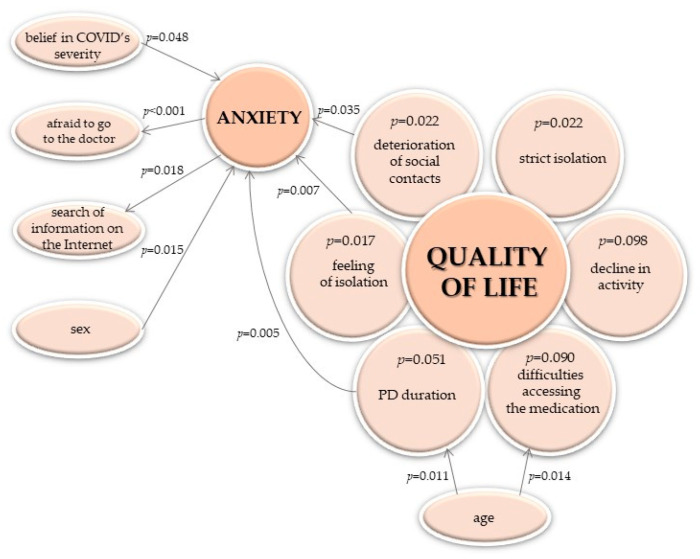
The chain of dependence between the change in the quality of life and other studied features. The presented *p*-values for quality of life refers to the frequency of its decline—results of the chi-square or the Fisher exact test, respectively (not to its change in points, per the Mann–Whitney test); the only exception is the association with activity reduction.

**Table 1 brainsci-12-00026-t001:** Characteristics of the study group (N = 47 respondents).

	Number of Cases	% of N *	95% Confidence Interval (CI)
Categorical variables			
Sex			
Men	30	64%	50–78%
Women	17	36%	22–50%
Education			
Tertiary	27	57%	43–72%
Secondary	13	28%	15–40%
Basic–vocational	7	15%	5–25%
Professional situation			
Full-time employees	4	9%	1–16%
Retired	43	91%	84–99%
The place of residence			
City over 500,000 inhabitants	35	74%	62–87%
City/town below 500,000 inhabitants	10	21%	10–33%
Village	2	4%	0–10%
Disease duration			
Up to 5 years	17	36%	22–50%
6–10 years	20	43%	28–57%
11 years or more	10	21%	10–33%
Independence in the performance of daily activities			
Complete independence	12	26%	13–38%
Little aid required	26	55%	41–70%
Complete dependence on others	9	19%	8–30%
Quantitative variable			
	Number of cases	Median	95% confidence interval (CI) for the median
Age	47	72.1 years	69.6–74.6 years

* Percentages do not have to add up to 100% due to rounding

**Table 2 brainsci-12-00026-t002:** Types of physical activities performed by people with PD during the pandemic.

Type of Activity	Number of Respondents	% of Respondents [N = 47]
Home training as before	26	55%
Walks	25	53%
Self-found exercises	15	32%
Individual exercises with a physiotherapist at home	7	15%
Individual exercises with physiotherapist online	7	15%
Cycling	7	15%
Team sports (e.g., boxing)	6	13%
Other activities	6	13%
None	3	6%

**Table 3 brainsci-12-00026-t003:** The deterioration of symptoms in the pandemic period reported by PD respondents.

Symptom	Number of Respondents	% of People Reporting Deteriorating Symptoms [n = 39]	% of All Respondents [N = 47]
Gait problems	12	31%	26%
Stiffness	10	26%	21%
Problems with daily activities	10	26%	21%
Slowness of movement (bradykinesia)	9	23%	19%
Problems with mood, motivation	8	21%	17%
Balance disorders	6	15%	13%
Problems with memory	6	15%	13%
Depression	4	10%	9%
None reported	8	21%	17%

## Data Availability

The complete questionnaire and its results are available in Appendix A.

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
