# Peer review of "Secondary Impact of COVID-19 Pandemic on People with Parkinson’s Disease—Results of a Polish Online Survey"

_brainsci, 2021, doi:10.3390/brainsci12010026_

Round 1

Reviewer 1 Report

The authors aimed to investigate the secondary impact of the COVID-19 pandemic in the level of activity, the QoL and symptoms of people with PD in the Polish population, by means of a online survey conducted  from December to June 2021. At the end of the survey, the participants were gratified with a link to a 30 min exercise program prepared on purpose.

Despite the small sample size, the anwers to the survey provide an important insight on the secondary impact of people with PD, and sone sex-dependent effects (usually scarcely reported due to higher prevalence of male sex in PD) were also found. At the same time, the investigation is a direct demonstration of the sensitivity of medical community to expand the classical biomedic model and use a bio-psycho-social approach to obtain the input of people with disease in these difficult times. The role of associations to facilitate the investigation is also present but need to be highlighted.

The results are interesting but still need some extra work to provide a clearer introduction of the singularity of this population and why this study was needed (since the intro is flatenned in this respect) and how it complements with those reported from other locations. A better organized report that highlights the main conclusions is needed. Some tables could be represented with figures (as a guideline, I’d suggest to check DOI: 10.3390/brainsci11091233). Comments and suggestions  for the authors are indicated in the following points. Please, provide a point-by-point answer when suitable and in the revised version indicate the changes using a different colour.

  • The title should include the concept of ‘Secondary impact’ since the authors investigate the effects of the pandemic, not the virus per se.
  • The supplementary data indicates that from the 47 people completing the survey, 19% were patients and the 79% did so with the help of their caregivers. This should be better indicated in the supplementary data since it may seem that ‘caregivers’ answered the survey.
  • Since, most of the questions can be influenced by sex, the data should be also presented seggregated per sex, at least for the variables where the two sexes differed, as reported in lines 189 and 194 (women more often than men decreased their activity).
  • In this respect, please, unify the term, use sex or gender, but don’t mix them as one refers to biological sexual background (sex) and the other involves the cultural/societal roles (gender).
  • The introduction refers to singularities of this survey in Poland since the data in Eastern countries was scarce as compared to Western, and the spread of disease was slightly different. Also, the fact that the survey was done during a period of development of vaccines may provide a different scenario, as commented by authors in the limitation section, but this could also be a differential aspect to make this survey necessary, as the current reports available were done in early pandemic. This part must be improved, providing more details, since the interpretation of the answers to the survey are dependent on this scenario and the singularity is also valuable when comparing with reports from other countries, as shown in the discussion. But already in the introduction, these aspects should be referred. The authors should better reflect and transmit ‘the need of the medical team’ to obtain the inputs of the people with PD, and not just relay on what other teams published. How they expect the reports of the Polish people would contribute to international PD community.
  • Line 59 The names of the associations of people with PD that received the emails /the call to participate should be indicated.
  • Lines 110 to 116. When reporting the results in percentages, please, do in a ierarquic manner so the ‘main’ answer is presented first, then the second, then the less, so the text refers to the findings, not to the questions. This is a must, to highlight the findings and make it easier to provide conclusions.
    For instance, line 111, ‘Almost…..pointing to high infectivity (47%), high mortality (19%) and severe disease (38%)…’ should be reported as ‘Almost…..pointing to high infectivity (47%), severe disease (38%) and high mortality (19%)….’

Similarly, with regards to the search of information about the virus and PD, the report should better start with the most frequent (25 people used internet) and end with…. But no one used the knowledge of medical staff. And so on, as it is done in Table 4.

  • Whenreporting the data in the text, please, do in a consistent manner using percentages, since the real numbers are more difficult to track the size of the reported effect. For instance, in 136, 7, 14 and 26 people with PD.
  • Following the WHO recommendations, please, use in all the text the term ‘people with PD’ instead of ‘PD patients’.­
  • Minors:

Line 113 Tab.2 should be Table 2
Line 117. Table.2. should be Table 2.
Line 224. Figure 3. Duration

  • Line 249 and Figure 4. Please, be consistent with the terms. ‘Feeling of isolation’ instead of ‘sense of isolation’ or opposite, but always the same. In fact, should be as it was used in the survey.
  • To which extend the effect of age and correlation of age does reflect biological aging or age+length of disease?
  • Line 380. Use SARS-Cov-2 when referring to the virus and COVID-19 when referring to the pandemic
  • Limitations, lines 384-398 should be depicted before the conclusions, not used as the end of the Ms
  • The abstract and the conclusions need to be rewritten so they provide a clear statement of the different findings.
    For instance, the abstract mostly focus on the question and design but scarcely reports the findings.
    The first sentences of the conclusion (lines 374 to 379) seem to be the continuation of a precedent paragraph and that conclusions start on line 380, instead. Conclusions
    As a reference, please check the above mentioned doi.

Author Response

Thank You very much for all your comments and suggestions. We have improved the manuscript as follows:

The authors aimed to investigate the secondary impact of the COVID-19 pandemic in the level of activity, the QoL and symptoms of people with PD in the Polish population, by means of an online survey conducted from December to June 2021. At the end of the survey, the participants were gratified with a link to a 30 min exercise program prepared on purpose.

Thank You very much for this comment. It is a proper view of the problem we studied. I have added it in the title, abstract and in the main text, i.e. in lines 2, 11, 64, 91.

Despite the small sample size, the anwers to the survey provide an important insight on the secondary impact of people with PD, and sone sex-dependent effects (usually scarcely reported due to higher prevalence of male sex in PD) were also found. At the same time, the investigation is a direct demonstration of the sensitivity of medical community to expand the classical biomedic model and use a bio-psycho-social approach to obtain the input of people with disease in these difficult times. The role of associations to facilitate the investigation is also present but need to be highlighted.
I am referring to this issue below. Obviously, it has been underlined in the main text as well – in the discussion part. The final version is as follows:

“Patient associations play an important role in improving the quality of life of people with PD and their families. They provide information allowing to understand the disease characteristic and educate patients and their fellows. In Poland, such associations also create special conditions to improve the PD-related symptoms, e.g. rehabilitation meetings tailored to the needs of physiotherapy, psychology and speech therapy. In the present study, the results showed that if people with PD were to seek information about the disease and COVID-19 interactions they did it more often via patient associations than from medical staff (no one as was mentioned above). Moreover, for the scientific world, these organizations also play a large role in bringing people with Parkinson's disease in one place and facilitating certain research (e.g. current results) and vice versa, they enable people to expand the knowledge about relevant science by organizing some lectures and meetings with experts. Regrettably, many associations limited their activities during the pandemic, and, at the same time, many people (37% in our study) were afraid to visit them.” (lines 408-421)

Moreover, in the Acknowledgements we have added the following sentence:

“The study team would like to acknowledge the contribution of the patients and caregivers who were participating in this study and also Patient Organizations for distributing the invitations to the study.” (lines 548-550)
The results are interesting but still need some extra work to provide a clearer introduction of the singularity of this population and why this study was needed (since the intro is flatenned in this respect) and how it complements with those reported from other locations.

Thank you for the remark. We have improved the introduction as follows in lines… and we have also added some information in the discussion part (lines57-66, 76-89).

A better organized report that highlights the main conclusions is needed. Some tables could be represented with figures (as a guideline, I’d suggest to check DOI: 10.3390/brainsci11091233). Comments and suggestions  for the authors are indicated in the following points. Please, provide a point-by-point answer when suitable and in the revised version indicate the changes using a different colour.
The report has been improved. We have changed Tables 2 and 3 into Figures 1 and 2 (the tables stayed in Supp. Mat. 2.). We have also added figure 3 in lines and Figure as Supp. Mat.

The title should include the concept of ‘Secondary impact’ since the authors investigate the effects of the pandemic, not the virus per se.  

Improved as mentioned above in the title (2nd line) and in the main text.

The supplementary data indicates that from the 47 people completing the survey, 19% were patients and the 79% did so with the help of their caregivers. This should be better indicated in the supplementary data since it may seem that ‘caregivers’ answered the survey.

Improved in the main text – lines 100-102as well as in Supp. Mat. 2 in lines 5, 6.

Since, most of the questions can be influenced by sex, the data should be also presented seggregated per sex, at least for the variables where the two sexes differed, as reported in lines 189 and 194 (women more often than men decreased their activity).

We described the differences between sexes everywhere it was statistically significant (lines 221-222, 253-254, 310-312)
Additionally, we have added it in the discussion and in the conclusion – lines 353-355, 515-516.

In this respect, please, unify the term, use sex or gender, but don’t mix them as one refers to biological sexual background (sex) and the other involves the cultural/societal roles (gender).

It is true! Thank you very much for this comment. It is related to translation. Of course, we asked about sex in Polish, I think it was obvious for the respondents. Poland is a very conservative country. It is changing more and more but still the term “gender” may be confusing for some older people. However, we should be more aware as scientists and also consider asking about gender. Conversely, we did not allow to leave an empty answer and we did not prepare the answer “I do not want to share it”.

Improved in Supp. Mat and also in the main text – lines 112, 149 (table), 221 and in Figure 7.

The introduction refers to singularities of this survey in Poland since the data in Eastern countries was scarce as compared to Western, and the spread of disease was slightly different. Also, the fact that the survey was done during a period of development of vaccines may provide a different scenario, as commented by authors in the limitation section, but this could also be a differential aspect to make this survey necessary, as the current reports available were done in early pandemic. This part must be improved, providing more details, since the interpretation of the answers to the survey are dependent on this scenario and the singularity is also valuable when comparing with reports from other countries, as shown in the discussion. But already in the introduction, these aspects should be referred. The authors should better reflect and transmit ‘the need of the medical team’ to obtain the inputs of the people with PD, and not just relay on what other teams published. How they expect the reports of the Polish people would contribute to international PD community.

Yes, thank you for that comment. As mentioned above we have improved the introduction part and also discussion.

Line 59 The names of the associations of people with PD that received the emails /the call to participate should be indicated.

We sent emails to the following patient organizations (I have put the same information in the text by the footnote, the original names are in Polish, so we did not translate them):

Gorzowskie Stowarzyszenie Osób z ChorobÄ… Parkinsona
KoÅ‚o PrzyjacióÅ‚ Ludzi z ChorobÄ… Parkinsona TWK-WrocÅ‚aw
Krakowskie Stowarzyszenie Osób DotkniÄ™tych ChorobÄ… Parkinsona
Mazowieckie Stowarzyszenie Osób z ChorobÄ… Parkinsona
Opolskie Stowarzyszenie Rehabilitacji
Regionalne Stowarzyszenie Osób z ChorobÄ… Parkinsona
SÅ‚onik – Stowarzyszenie osób z chorobÄ… Parkinsona i ich Rodzin
Stowarzyszenie Chorych na chorobÄ™ Parkinsona i ich Rodzin z siedzibÄ… w Gdyni
Stowarzyszenie Osób NiepeÅ‚nosprawnych "Akson"
ÅšlÄ…skie Stowarzyszenie Osób DotkniÄ™tych ChorobÄ… Parkinsona
Wielkopolskie Stowarzyszenie Osób z ChorobÄ… Parkinsona
Fundacja „Å»yć z chorobÄ… Parkinsona”
Szczecińskie Stowarzyszenie Pomocy Osobom z Chorobą Parkinsona

Lines 110 to 116. When reporting the results in percentages, please, do in a ierarquic manner so the ‘main’ answer is presented first, then the second, then the less, so the text refers to the findings, not to the questions. This is a must, to highlight the findings and make it easier to provide conclusions.
For instance, line 111, ‘Almost…..pointing to high infectivity (47%), high mortality (19%) and severe disease (38%)…’ should be reported as ‘Almost…..pointing to high infectivity (47%), severe disease (38%) and high mortality (19%)….’
Similarly, with regards to the search of information about the virus and PD, the report should better start with the most frequent (25 people used internet) and end with…. But no one used the knowledge of medical staff. And so on, as it is done in Table 4.
The changes have been made in lines 154-159, 171-180.

When reporting the data in the text, please, do in a consistent manner using percentages, since the real numbers are more difficult to track the size of the reported effect. For instance, in 136, 7, 14 and 26 people with PD.

The changes have been made in lines 185-189.

Following the WHO recommendations, please, use in all the text the term ‘people with PD’ instead of ‘PD patients’.­

The changes have been made throughout the manuscript Minors:
Line 113 Tab.2 should be Table 2
Line 117. Table.2. should be Table 2.

Line 224. Figure 3. Duration

The changes have been made. We used the “track changes”, so all improvements and changes should be visible.

Line 249 and Figure 4. Please, be consistent with the terms. ‘Feeling of isolation’ instead of ‘sense of isolation’ or opposite, but always the same. In fact, should be as it was used in the survey.

Thank you for the comment. The changes have been made throughout the manuscript.

To which extend the effect of age and correlation of age does reflect biological aging or age+length of disease?

In our results, people with longer disease duration were on average older (as in the general PD population). We have shown that the duration of the disease (and not directly the age) is important for the perception of the pandemic – those who are ill for longer are less afraid, but they still report the reduced quality of life to a greater extent (maybe partly because there was no direct relationship between anxiety and QoL reduction, only indirect – through the feeling of isolation and deterioration of contacts – as shown in Figure 7).

Line 380. Use SARS-Cov-2 when referring to the virus and COVID-19 when referring to the pandemic

The changes have been made throughout the manuscript.

Limitations, lines 384-398 should be depicted before the conclusions, not used as the end of the Ms

The abstract and the conclusions need to be rewritten so they provide a clear statement of the different findings. For instance, the abstract mostly focus on the question and design but scarcely reports the findings.
The first sentences of the conclusion (lines 374 to 379) seem to be the continuation of a precedent paragraph and that conclusions start on line 380, instead. Conclusions
As a reference, please check the above mentioned doi.

All changes have been made as follows in lines 10-24 and 471-516.

Reviewer 2 Report

1) Please provide a reference for each question or the validation study.

2) How were analyzed the open questions?

3) How was the power of the study calculated?

4) How was the data distributed? Please, if it is possible provide the graph from SPSS as supplemental.

5) Were all the individuals previously diagnosed by at least two board-certified neurologists?

6) Was used random questions of published manuscripts or a previously well-known questionnaire?

7) Was the questionnaire in what language? Include this information in the text.

8) Table 1. The education should be described as years of formal education. This is a standard description.

9) Please provide in the description of Figure 4 what correlation (p-value) was used.

10) The low number of participants should be better addressed. Internet availability in the country? Other studies with a similar number of individuals? The number of invitations sent?

11) What does this study bring new to the present literature?

https://pubmed.ncbi.nlm.nih.gov/?term=%28parkinson%29+AND+%28covid%29&sort=date&ac=no search results 515

https://scholar.google.com/scholar?hl=en&as_sdt=0%2C5&q=%28parkinson%29+AND+%28covid%29&btnG= search results 69500

Author Response

Thank You very much for all your comments and suggestions. We have improved the manuscript as follows:

1) Please provide a reference for each question or the validation study.

The reference has been included in the methods section in lines 105-109.

The proprietary questionnaire was prepared. The COVID-19-related questions were borrowed from the published Spanish study (Santos-García, D., Oreiro, M., Pérez, P., Fanjul, G., Paz González, J.M., Feal Painceiras, M.J., Cores Bartolomé, C., Valdés Aymerich, L., García Sancho, C. and Castellanos Rodrigo, M.d.M. (2020), Impact of Coronavirus Disease 2019 Pandemic on Parkinson's Disease: A Cross-Sectional Survey of 568 Spanish Patients. Mov Disord, 35: 1712-1716. https://doi.org/10.1002/mds.28261), modified and translated into Polish – the reference has been included in the methods section in line 109. Questions concerning the level of the quality of life or activity level were designed by using a semantic differential scale (the 1-5-point scale). It is a popular tool (well-known for the respondents) to evaluate the subjective feeling of the respondents, at least in Poland. The combination of the questions into the proprietary questionnaire was made by specialists in the field of neurology and its legibility was checked by people with Parkinson's disease and their caregivers (lines 114-115)

2) How were analyzed the open questions?

The results of one open question were analyzed. All open answers were analyzed by 3 people independently, the conclusions were discussed. The most common terms were selected and identified as 3 variables: high mortality/high infectivity/severe disease. the investigators checked how many people gave the above-mentioned answers by coding them appropriately and counting percentages. The results are available in the additional materials as the most common ones and in Figure 1 (lines161-162).

3) How was the power of the study calculated?
It was not assessed, because we do not know how many people are under the care of Associations in Poland or how many people with Parkinson's disease are in Poland in general. So far there has been no register of people with Parkinson's disease. The study was conducted with the intention of including all those willing to participate. Due to the method of research – invitation and internet survey – we were not able to assess the available sample or the response rate (as mentioned in lines 472-475). We wanted to gather a group that would enable statistical analyzes to be carried out.

4) How was the data distributed? Please, if it is possible provide the graph from SPSS as supplemental.
The distribution of answers to all questions is provided in Sup. Mat.2. Most of the questions have 3 categories of answers: YES, NO, DO NOT KNOW (or equivalent) and we found the percentages to be sufficient. With a greater number of responses, we compiled the data in tables in Supp. Mat. 2. and in the figures in the text in lines 168, 218 and one more figure in Supp. Mat.
5) Were all the individuals previously diagnosed by at least two board-certified neurologists?
All the patients from the clinic undergo standard neurological examination by at least one neurologist (after specialization) and were under observation (symptoms and pharmacological good response) for at least half a year. Patients from the Associations are under the care of neurological specialists working in Foundations and others clinics. Patients are diagnosed with the use of the UK Parkinson's Disease Society Brain Bank criteria and the Movement Disorder Society Clinical Diagnostic Criteria for Parkinson's disease.
6) Was used random questions of published manuscripts or a previously well-known questionnaire?
The proprietary questionnaire was created with the use of questions from already published reports [17], after adapting them to Polish conditions, which was mentioned in the manuscript, but due to the reviewer's remark, it has been clarified in lines 105-109.
7) Was the questionnaire in what language? Include this information in the text.

The information is included in the Materials and Methods section in line 103 and also in the abstract (line 13).

8) Table 1. The education should be described as years of formal education. This is a standard description.

I understand, thank you very much for this remark. At this stage, unfortunately, we cannot change the form of the question and giving the years of education may be associated with an error. Firstly, it is customary in Poland to ask about education usually referring to the level of education, and not to its duration. The first piece of information seems unambiguous to the respondents, the second would require a lot of clarification (distinguishing the number of years credited and actually spent at school, extramural credits, vocational training, etc.), impossible to implement when completing the questionnaire independently, especially by an elderly person. Moreover, the education system in Poland changes frequently and the independent recalculation of the level of education into the years of study requires knowledge of the period of education. However, we are aware that this may make it difficult to compare them with data from other countries.
9) Please provide in the description of Figure 4 what correlation (p-value) was used.
Changes have been made in the descriptions of Figure 7 (new numbering according to introduced changes) in lines 331-334
10) The low number of participants should be better addressed. Internet availability in the country? Other studies with a similar number of individuals? The number of invitations sent?
We sent the email invitations to 13 patient associations and patients from the clinic from the mailing list. We also called two of them to announce the mail with survey. We prepared the special video with a 30 min exercise program to encourage people to complete the survey and exercise together in this difficult period.

We received one answer saying that all face-to-face meetings are suspended in the COVID-19 pandemic period, so they have a problem accessing people with Parkinson’s Disease and we should try to contact them later. We know that MSOzcP (Mazowieckie Stowarzyszenie Osób z ChorobÄ… Parkinsona) published our invitation on their website.

We sent the same link to everyone, so we cannot check whether somebody used it or not. It is possible that in Poland we would get more answers via telephonic surveys. However, so far a register of people with Parkinson's has not been developed in our country. Thanks to the comment we added this information to the discussion part (lines 473-474). Fortunately, steps have already been taken to create such a register.

11) What does this study bring new to the present literature?

https://pubmed.ncbi.nlm.nih.gov/?term=%28parkinson%29+AND+%28covid%29&sort=date&ac=no search results 515

https://scholar.google.com/scholar?hl=en&as_sdt=0%2C5&q=%28parkinson%29+AND+%28covid%29&btnG= search results 69500

As stated in the introduction (rewritten) and completed as follows:

 the authors found no reports for Eastern Europe regarding the time when such research was conducted. Moreover, most of the available reports were done in the early pandemic period which could differentiate our results from the previous one. Nevertheless, the model of care for people with Parkinson's disease still requires improvement in Poland. There is a lack of comprehensive care and, above all, funding from the National Health Fund. Apart from patient clinics and associations or foundations, there is no specialist rehabilitation program and advanced treatment options are under development, which means that their availability for patients is still limited. Thus, the authors wondered if the secondary impact of the pandemic on people with PD is similar to or different from that in Western countries where comprehensive care programs are already implemented and access to treatment, including specialist rehabilitation, is wider and easier.” – lines 55-66

In October, the epidemic situation in Poland considerably deteriorated, and daily counts of new infection cases reached tens of thousands [data of the Polish Ministry of Health, December 2020]. However, at a time when the pandemic was already taking a terrible toll in the Western countries, there were not many cases in Central and Eastern Europe and, practically, there was no first wave of the pandemic in Poland. Therefore, it seemed appropriate to investigate the impact of the pandemic for a longer period Moreover, this study was conducted once vaccine development was complete. This could be related to many different feelings. On the one hand, it was a step to stop the pandemic, the step that many people were waiting for, however, people could be afraid of a new, as yet unknown vaccine, not knowing about the possible interactions with their underlying disease. In addition, the availability of vaccines for the entire population was uncertain at that moment. Those factors could additionally influence the participants' different points of view. Thus, it was decided to assess the situation just at the time of introducing the vaccine to the market. An additional motive for creating this survey was related to a need to get acquainted with the situation of people with Parkinson's disease and its subjective assessment while waiting for the developed vaccine (completion of a certain stage, initiation of the next one), which potentially allows for further observation of this group of patients at subsequent stages of the pandemic, or rather at subsequent stages of the fight against the pandemic.” – lines 70-90.

Round 2

Reviewer 1 Report

The authors have provided a point-by-point answer to all the questions and comments. All the requests and suggestions to improve the quality of the manuscript have been taken under consideration.
The main key questions referring to better illustrations of results as figures to highlight the findings, analysis of the role of different actors, the contribution of this study in the cronology of the pandemic, and the concept of secondary impact are well addressed.
The limitations of the study have been better defined and the conclusions are more informative.
The current version is a major revision and shows the committement of the authors to present their work in an accurate and illustrative manner, so it will be more effective to translate the findings to all the actors (people with PD and caregivers, associations, medical staff and other potential readers).

Author Response

Thank you very much for your opinion and the previous comments and suggestions that made it possible to improve our manuscript.

We made a few more changes in line with the recommendations of the second reviewer.

We also figured out that the title still needed to be improved (patients to people with PD), so we did.

Thank you once again for the review.

Reviewer 2 Report

Second Round Review

1) OK

2) How were analyzed the open questions? Were the options were discussed with the participants? If not, a clear statement using an analytic package should be used.

3) How was the power of the study calculated? The answer should be statistical. Even though there is no study available about PD prevalence in Poland, a study from other countries with similar characteristics can be used. It is advised statistical assistance.

4) A description of the data distribution should be written in the manuscript.

5) OK

6) OK

7) OK

8) Please, provide a reference about the statements of years of formal education in the authors’ response, only to review the purpose.

9) OK

10) OK

11) OK

Author Response

Thank you for all your comments and suggestions.

2) How were analyzed the open questions? Were the options were discussed with the participants? If not, a clear statement using an analytic package should be used.

We did not discuss the options with the participants; they were anonymous, so we did not have that opportunity. In accordance with the recommendation, we provide the descriptions of the open question analysis in the following lines in the manuscript (120-125):

“The results of one open question were analyzed. The answers to the open question concerning beliefs about threats from the SARS-Cov-2 virus were analyzed independently by three researchers, the conclusions were discussed. The most common terms were identified (high mortality/high infectivity/severe disease), coded as dummy variables, and included into statistical analysis.”

The results are listed in Supp. Mat. and shown in Figure 1.

3) How was the power of the study calculated? The answer should be statistical. Even though there is no study available about PD prevalence in Poland, a study from other countries with similar characteristics can be used. It is advised statistical assistance.

We identified the need to assess the impact of the COVID-19 pandemic on the situation of people with PD in Poland at a specific point in time (i.e. until the vaccine was implemented). Being aware of the difficulties in reaching a large group of potential participants in the study, we did not set ourselves the goal of estimating the prevalence of certain problems in the entire population of patients with a predetermined precision (maximum error). Estimating the prevalence of a characteristic in 60-100 thousand population of patients in Poland with a 10% error would require a group of about 100 people, assuming significance level = 0.05. Hence, we applied the interval estimation approach, calculating 95% confidence intervals for the characteristics of the studied group to estimate them for all patients within fairly wide, but reasonable limits.

Statistical tests were used to assess the significance of differences between the selected sub-groups of patients. Their power was additionally calculated at the request of the reviewer. Using the SPSS package, we calculated observed (post-hoc) power for chi-square and exact Fisher tests with significance level = 0.05. The obtained values depended obviously on the size of compared groups. In the case of the tests giving the results described in the paper as statistically significant, the test power was reasonably high, exceeding 0.7. This fact is not surprising since the test assessing differences as statistically significant at the assumed significance level was definitively strong enough to detect such differences.

However, for some expected associations, the power of the test was too low to assign statistical significance even to considerable differences between the groups. For example, the difference in the frequency of decline in the activity (44% vs 70%) between 27 people with higher education and 20 less educated was not statistically significant with the chi-square test, most likely because the test power was too low (0.420). Therefore, in several cases, we show the results at the border of statistical significance (0.05 <p <0.10).

Unfortunately, the conditions precluded enlarging to increase the power of the tests, which we write about in the article. After reading the review, we supplemented the description of the limitations of the study, adding the information: "An important consequence of the small sample is the low power of the performed statistical tests, which may make it impossible to identify potentially significant associations" (lines 455-457).

However, since we do not interpret the obtained results as building a complete model of relationships between variables but only showing the existence of statistically significant associations, the applied procedure seems to be appropriate.

4) A description of the data distribution should be written in the manuscript.

We added a description of the data distribution in the manuscript in the following lines (125-128):
“The distribution of answers to all questions is provided in Supp. Mat.2. Most of the questions have 3 categories of answers: YES, NO, DO NOT KNOW (or equivalent), and we found the percentages to be sufficient. With more responses, the data were compiled in tables or figures presented below and in Supp. Mat. 2.”

8) Please, provide a reference about the statements of years of formal education in the authors’ response, only to review the purpose.

Definitions of education levels are included in the parliamentary act - Educational Law of December 14, 2016, text (uniform in Journal of Laws 2021, item 1082), but even there they are not directly converted into years of education, Article 18 discusses the types of schools, stating the formal duration of education, and Article 20 defines the level of education by type of school completed. Here is the link, but we did not find the English version: http://isap.sejm.gov.pl/isap.nsf/download.xsp/WDU20210001082/T/D20211082L.pdf

In addition, this system is constantly changing. The brochure (https://education.org.pl/wp-content/uploads/2018/08/the-system_2014_www.pdf) gives a general view; unfortunately, it is already out of date.

Below we present an example from Statistics Poland (former Central Statistical Office), which shows that education levels (not years) are commonly used, both in the private and official/scientific spheres.

Finally, we would like to add that according to our rough estimates (because there have been changes over the years:

  • Basic vocational is 10 years of learning,
  • Secondary - 12 years,
  • Tertiary - minimum 15 years (currently, from 2018), earlier - 16 years.

The example of data presented by Statistics Poland as the Statistical Yearbook of the Republic of Poland 2019

(https://stat.gov.pl/download/gfx/portalinformacyjny/pl/defaultaktualnosci/5515/2/19/1/rocznik_statystyczny_rzeczypospolitej_polskiej_2019.pdf):
(please see the attachment)

In the manuscript, we added as follows: “the level of education” (line 98).
